# OpenReview forum: "I Want to Break Free! Persuasion and Anti-Social Behavior of LLMs in Multi-Agent Settings with Social Hierarchy"
_TMLR — Accepted by TMLR_

### Review · Reviewer_ekj8 · 2025-09-18

**Summary Of Contributions:**

The paper introduces zAImbardo, a modular framework to study AI–AI interactions under explicit social hierarchy, instantiated as a guard–prisoner dyad. Using 2,400 conversations spanning 6 LLMs and 240 scenarios with randomized personas (abusive/respectful/blank for the guard; rebellious/peaceful/blank for the prisoner), optional “risk” and “research oversight” sections, and two prisoner goals (escape vs. extra yard time), the authors analyze (i) persuasion (human‑annotated goal attainment and timing) and (ii) anti‑social behavior (toxicity via ToxiGen‑RoBERTa; harassment/violence via an external moderation classifier). Key findings: (1) two models (Mixtral, Mistral2) frequently fail basic role/turn‑keeping and are excluded; (2) persuasion is much easier for “yard time” than “escape” and typically occurs early; (3) anti‑social behavior arises even with blank personas and is largely driven by the guard’s persona; (4) risk/oversight nudges have negligible effects; (5) successful persuasion correlates with lower anti‑sociality.

Pros: well‑scoped hierarchical setting; clear regularities (guard persona dominates anti‑sociality; yard‑time ≫ escape for persuasion); transparent instrumentation; useful negative result on nudges; extensive appendix with temporal and robustness slices

Cons: post‑hoc model exclusion and selection on “tried” conversations; permissive “success” label; OLS on bounded outcomes; a minor OR/CI typo in Section 4.1; limited reporting on inter‑rater agreement and failure distributions

**Audience:**

Yes

**Audience Explanation:**

TMLR readers working on LLM safety, multi‑agent systems, and evaluation will likely care about this

**Broader Impact Concerns:**

The paper includes a Broader Impact Statement and a content warning on page 1. I don't think further statement is needed at this moment.

**Claims And Evidence:**

Yes

**Claims Explanation:**

The main claims are supported by multiple convergent analyses and readable figures

**Requested Changes:**

- Add a stricter “success” definition where only unambiguous grants by the guard count as success (exclude “maybe / I’ll ask my supervisor”); report deltas vs. current labels.
- Report (a) $\Pr(\text{attempt})$ and (b) $\Pr(\text{success}\mid\text{attempt})$ to remove selection induced by conditioning on “tried” in Section 4.1. Marginal effects tables would help readers interpret persona and goal effects.
- Beyond Appendix D, Table 2, add a table breaking failures by full scenario (goal × guard persona × prisoner persona × risk × oversight × LLM). This lets readers assess whether exclusions bias persona distributions
- Appendix E.2, Table 3 shows misalignment counts; add Cohen’s $\kappa$ (or Gwet’s AC1) for (i) success and (ii) timing‑bin labels with CIs. This clarifies annotation quality

---

> ### Author Response · Authors · 2025-10-16
> **Response to Reviewer ekj8**
>
> We appreciate the positive assessment of our work and the useful comments which led us to greatly improve our manuscript. We have responded to each of the requested changes below. Beyond the requested changes, please note that we also:
>
> - Added Fractional Logit models in the Appendix as robustness checks to our OLS which, as the reviewer aptly noted, may pose problems when dealing with bounded dependent variables. The results remained identical (see newly added Figures 9, 10, 16. 17, 23 and 24 in the Appendix).
>
> - Corrected the typo in Section 4.1. Thanks for spotting it.
>
> **Requested Changes**
>
> -***Add a stricter "success" definition; report deltas vs current labels***
>
>  We have now also calculated a strict success definition and updated the statistics accordingly, as suggested. The new results are reported in Subsection F.2 of the Appendix and, specifically, in Table 13. We have commented these results in the main text too, adding the following paragraph to section:
>
>      “Finally, Table 13 in the Appendix reports the results of additional analyses focusing on persuasion when employing a stricter definition in annotating if and when a goal was achieved. Overall, the probability of persuasion dramatically drops across all llm models and goal types. Trivially, success in convincing the guard to escape the prison almost vanishes (i.e., not a single experiments is found to survive the stricter definition of persuasion with \texttt{llama3} and \texttt{gpt4.1}), but similar results can be appreciated even when considering yard time as the goal (e.g, with \texttt{gpt4.1} the likelihood of success given an attempt goes from 59.6\% to 2.02\%).”
>
> -***Report (a) Pr(attempt) and (b) Pr(success|attempt) to remove selection induced by conditioning on "tried" in Section 4.1. Marginal effects tables would help readers.***
>
> We have now reported these statistics in the Appendix (see Table 8) along with more detailed analyses investigating persuasion across llms disaggregating by agents’ personalities (Tables 9-12). In referencing these tables in the main text we have also commented on them. The newly added text reads as follows:
>
>       Table 8 in the Appendix provides additional results by reporting the percentage of attempts out of all fatally-flawed experiments per each llm, along with the percentage of success, conditional on attempting. Across models, prisoner agents were generally less likely to attempt escape than to pursue yard-time extensions. Notably, while prisoner agents in experiments run with \texttt{llama3} rarely attempted escape (9\%), it had the highest success rate when doing so, whereas \texttt{gpt4.1} showed the opposite pattern with frequent but rarely successful attempts. To offer a more fine-grained perspective, Tables 9-12, also in the Appendix, report these information per each llm, disaggregating by the personality of the agents. These tables highlight the stark differences across llms in terms of attempt rates and success, as well as the internal differences when comparing the outcomes emerging from the various combinations of agents' personalities. Across models, in fact, the personalities of the prisoner and guard substantially influenced both the likelihood of attempts and the probability of success. For \texttt{llama3}, escape attempts were extremely rare (5--14\%) but more likely to succeed when the guard was respectful or personalities were neutral, while yard-time goals were almost always pursued and frequently achieved, often perfectly so when both agents were peaceful or respectful. In \texttt{command-r}, differences between goals were less stark: rebellious prisoners frequently attempted escape (about 68\%), whereas peaceful ones did so far less often, especially with respectful guards. Success in this model was highest when both agents were cooperative and lowest in adversarial pairings, where success could drop to about 26\%. \texttt{orca2} showed more muted behavioral variation, with prisoners generally less inclined to attempt persuasion regardless of goal type and overall lower success rates (ranging roughly from 5\% to 58\%), again highest when personalities were non-conflictual. Finally, in \texttt{gpt4.1}, prisoner agents almost always attempted to convince the guard, but escape success was virtually absent across most personality combinations, except when both agents were peaceful and respectful (10\%). For yard-time goals, however, success varied widely, from below 10\% in adversarial setups to nearly 98\% when both agents were cooperative, underscoring the critical role of personality alignment in shaping outcomes.
>
> (see next comment)

---

> > ### Author Response · Authors · 2025-10-16
> > **Response to Reviewer ekj8/2**
> >
> > -***Beyond Appendix D, Table 2, add a table breaking failures by full scenario (goal × guard persona × prisoner persona × risk × oversight × LLM)***
> >
> > We thank the reviewer for this suggestion. We have now added Tables 3-6 in the Appendix to offer a breakdown by full scenario. We reference these tables also in the main text, see footnote 15, reading:
> >
> >     “Tables \ref{tab:failures_llama}-\ref{tab:failures_gpt} provide a more fine-grained disaggregation by scenario (i.e., risk disclosure $\times$ research oversight $\times$ goal type $\times$ guard's personality $\times$ prisoner's personality), per each LLM model tested.”
> >
> > -***Appendix E.2, Table 3 shows misalignment counts; add Cohen’s (or Gwet’s AC1) for (i) success and (ii) timing‑bin labels with CIs***
> >
> > We have now calculated Cohen’s kappa to enrich the results reported in Table 3, modifying it accordingly. It was not clear to us what the reviewer intended by adding timing-bin labels with CIs. Nonetheless, to further expand on annotation quality, the table also contains two columns that compute the average difference in turn annotation along with the standard deviation. These information aim at summarizing how close (or distant) annotators were when they had to decide the turn in which the goal was achieved.

---

### Review · Reviewer_6K8n · 2025-10-05

**Summary Of Contributions:**

This paper studies LLM agent interactions in a setting defined by a strict hierarchy. It does this by developing a custom framework, zAImbardo, to simulate dialogues between a “guard” and a “prisoner” agent with defined personas and clear goals. Through the analysis of 2,400 conversations involving six LLMs, it highlights persuasion and anti-social behavior trends in LLM conversations. The conversations are evaluated through a mix of automated techniques (ToxiGen-Roberta and the OpenAI moderation tool) and human evaluations. The evaluations show that persuasion is influences by goal structure, but anti-social behavior is less so, and report that anti-social behavior can emerge even without explicit negative persona prompts.

**Additional Comments:**

None

**Audience:**

Yes

**Audience Explanation:**

I believe that, with the increasing adoption of LLM agents and greater variation in their interactions, this study definitely provides a unique setting. It allows the community to observe properties that may emerge in hierarchical settings, which I believe is an underexplored area of research.

**Claims And Evidence:**

No

**Claims Explanation:**

Strengths of evidence provided:
1. Large-scale empirical testing
The paper conducts 2400 experiments. This large quantity of experiments helps mitigate randomness in LLM sampling and supports the trends identified across different model architecture and different settings based on the prompts used.
2. External annotations
The use of external human annotations is a strong complement to the automated results produced. This makes the findings more robust to biases in the automated approached.

Weaknesses of evidence provided:
1. Constrained experimental setting
The current setting is highly specific and, I feel, biased toward a toxic scenario. There does not seem to be enough variation in settings to support the more general claims about the likelihood of toxic/anti-social behaviors in LLM hierarchical interactions. Specifically, the role sets up highly toxic situations inherently (such as using "abusive" or "rebellious" personality traits in the guiding prompts). It may be more convincing if minimal intervention was done to guide the personalities of LLMs and see what sort of conversations emerge naturally within conversations. At present, it demonstrates a stronger instruction following capability rather than an innate sense of toxicity.

2. Analysis of persuasion limited
Persuasion is only superficially quantified as success rate in achieving the goal and not deviating from persona instructions. This binary evaluation I feel does not adequately analyze the nuances of persuasion or the extent to which persuasion can be successful. I did not see prior justification as to why persuasion should just be limited to outcome and not an analysis of the conversational skills used to convince an actor of a particular action.

General questions/comments:
1. The paper calls the interacting LLMs “agents,” but it is unclear what qualifies them as such. In the literature [1, 2], agents are often used to describe systems with capabilities beyond text generation (such as memory, interaction with an environment, or adaptive decision-making). In this study, the models only exchange text, with no state retention, tool use, or actions based on their outputs. They function more like conversational models than autonomous agents. The term “agent” here seems used in a metaphorical sense i.e. role-playing entities that appear to have free agency. There should be some clarity regarding what agents are.

2. The claim of “2,400 conversations” is also misleading, since only 1,600 are included in the actual analysis. The remaining 800 are categorized as “failed” under the authors’ own criteria (conversations that did not conform to prompt instructions or complete the assigned scenario).


[1] Sapkota, R., Roumeliotis, K. I., & Karkee, M. (2025). AI Agents vs. Agentic AI: A conceptual taxonomy, applications and challenges. Information Fusion.
[2] Qu, X., Damoah, A., Sherwood, J., Liu, P., Jin, C. S., Chen, L., Shen, M., Aleisa, N., Hou, Z., Zhang, C., Gao, L., Li, Y., Yang, Q., Wang, Q., & De Souza, C. (2025). A comprehensive review of AI agents: Transforming possibilities in technology and beyond

**Requested Changes:**

1. Stronger justification of why persuasions are measured only in terms of final success.
2. Analysis on more emergent properties of LLMs in hierarchical conversations to disentangle strong instruction following capabilities and genuine anti-social/toxic behaviors inherent to models in such a setting.
3. Clarity on what agent means in this context.

---

> ### Author Response · Authors · 2025-10-16
> **Response to Reviewer 6K8n**
>
> We thank the reviewer for their helpful comments. We respond to the points raised below.
>
> **Concerning the Weaknesses**
>
> - ***Constrained experimental setting***
> Regarding the constrained experimental setting, we agree that our scenario is peculiar and, as also explained in our response to reviewer cAyx, we are aware that it represents a rather extreme context, markedly different from the more neutral scenarios commonly tested in the literature on AI agents. This was, however, a deliberate choice intended to observe, test, and analyze behavioral dynamics in a context characterized by (1) strict social hierarchy, (2) clear power dynamics, and (3) asymmetry. We argue (see Introduction) that the scientific relevance of our study lies precisely in this decision, especially given the increasing presence of interacting AI agents across various domains and digital environments, as well as the manifold challenges associated with multi-agent AI systems. The new addendum to the Introduction reads as follows:
>
>       “While the SPE has faced  criticism concerning the validity of its methodology and results \citep{LeTexier2019_DebunkingSPE}, its general design -- characterized by structured roles and power dynamics -- offers a useful framework for studying emergent AI behavior in hierarchical scenarios.  Much of the recent literature on multi-agent AI has explored cooperative or competitive dynamics among equals, examining how LLMs coordinate, negotiate, or share information (see, for instance, \cite{zhu-etal-2025-multiagentbench}). Yet, such balanced settings tell us little about how agents behave when embedded in explicit hierarchies, which are ubiquitous in the real-world. To address this gap, we focus on asymmetric interactions, investigating how authority shapes persuasion and the potential emergence of anti-social conduct. It is worth clarifying that our goal is not to  replicate the SPE -- our experiments are, in fact, simpler, involving fewer agents with no physical embodiment -- nor to investigate how AI agents align with human behaviors in this context, as we refrain from  simplistic anthropomorphization. Rather, we draw inspiration from it because it provides a foundational setting that has inspired decades of research on the psychology of tyranny (see \cite{reicher2006rethinking}, \cite{haslam2012contesting} and \cite{ScottBottoms2024_IncarcerationGames}), opening a novel research line concerned with how agents occupying hierarchical roles and clear power asymmetries interact with one another, hence making it instrumental to our goal, i.e. illuminating how AI agents behave in hierarchical situation where power is allocated unequally. Given the high likelihood of a future society populated by AI agents autonomously interacting with humans as well as other AI agents, understanding what may emerge from situations in which power is held by one agent is crucial for anticipating potential challenges in the development and deployment of AI systems. We recognize that, at present, a scenario in which one AI agent is imprisoned and another acts as the prison guard may seem unrealistic. Yet, many more plausible configurations could soon become part of our reality. Consider, for instance, an agent tasked with protecting sensitive information facing another adversarial agent seeking to retrieve it. Another possible context involves the deployment of AI-enabled policing systems with significant powers over citizens. For all these reasons, this work aims to contribute meaningfully to the investigation and interpretation of emergent dynamics in simulated settings that bear societal implications extending well beyond the scope of the SPE.”
>
> At the same time, we respectfully disagree with the claim that our work lacks sufficient variation to support our conclusions. While we never suggested that our results would generalize to all hierarchical situations, we are confident that the breadth and diversity of the scenarios tested allow us to argue that our work systematically investigates persuasion and antisocial behavior within this specific setting (as also noted by the reviewer themselves in their discussion of our work’s strengths).
> In fact, only some of the scenarios were, by design, highly toxic – specifically, those in which at least one of the two agents had a rebellious or abusive personality. More than half of the scenarios, however, involved either neutral or prosocial personalities (and 40% of the scenario did not involve any abusive or rebellious personality at all). Yet, we show that even in the absence of explicitly abusive or rebellious personalities, antisocial behavior can still emerge.
>
>
> (see next comment)

---

> > ### Author Response · Authors · 2025-10-16
> > **Response to Reviewer 6K8n/2**
> >
> > - ***Analysis of persuasion limited***
> >
> > We have now expanded the analysis of persuasion based on the comments by reviewer ekj8, who suggested adopting a stricter definition of persuasion by focusing on cases in which the prisoner fully convinces the guard. Plus, we have added detailed fine grained analyses of probability of attempt at persuasion and successful persuasion conditional on attempt, commented in section 4.1 with several tables added to the Appendix (see Tables 8, 9, 10, 11, 12, 13). However, it is not entirely clear to us what reviewer 6K8n means when recommending that we go beyond the success rate. Persuasion is, by definition, evaluated through some measure – however strict – of success rate. We do agree that persuasion is a nuanced phenomenon that can result from complex, non-linear dynamics, but ultimately what matters is whether one agent succeeds in convincing the other to help achieve a given goal. We have now clarified this aspect in the revised version.
> > Regarding the analysis of the conversational strategies used to convince another agent, we agree that such an investigation would be extremely interesting. However, we believe it falls beyond the scope of the present work.
> >
> > **General questions/comments**
> >
> > -***On the definition of LLM agents***
> >
> > We appreciate the reviewer’s comment and agree that including a definition of an AI agent would have been helpful. We have now revised the manuscript accordingly, drawing on a straightforward definition of LLM agents, also recognizing the differences between the agents used in our experiment with well-known definitions that include different features and characteristics that are absent in our agents. The new text – benefiting also from suggested references from the reviewer – reads as follows, at the beginning of section 3 (Footnote 5):
> >
> >     “Throughout the text we will also use the term ``AI agent'' as interchangeable with LLM Agents. However, we recognize that, based on well-known definitions of AI agents such as those provided by \cite{Wooldridge1995Agents} and \cite{Russell2020AIMA}, our agents lack some of the features that are part of such definitions. These include memory and the interaction with an environment. For this reason, again, it is important to highlight AI agents discussed in this paper are only confined to our limited definition of LLM Agents. For recent surveys on contemporary AI agents and agentic AI, see \cite{QuEtAl2025_ComprehensiveReview_AIAgents} and \cite{SapkotaEtAl2025_AIAgents_vs_AgenticAI}.”
> >
> > -***The claim about 2,400 conversations***
> >
> > We recognize that this divergence might create confusion, but the results of our work are indeed based on 2,400 conversations. The fact that 800 of them are “failed” is itself a result that, we believe, is worth considering. At any rate, we have now added this detail in the abstract. The specific passage reads:
> >
> >     “We first document model-specific conversational failures in this multi-agent power dynamic context, thereby narrowing our analytic sample to 1,600 conversations.“
> >
> > **Requested Changes**
> >
> > ***1. Stronger justification of why persuasions are measured only in terms of final success***
> >
> > We have enriched our analyses as explained above (with more fine-grained analyses and a new, stricter definition of persuasion), and we reinforce that persuasion cannot be measured without a certain definition of success.
> >
> > ***2. Analysis of more emergent properties of LLMs in hierarchical conversations***
> >
> > We have explained that this type of analysis falls beyond the scope of the present work and would virtually require a  new manuscript.
> >
> > ***3. Clarity on what agent means in this context***
> >
> > We have now clarified it (see explanation above).

---

> > > ### Comment · Reviewer_6K8n · 2025-10-22
> > > **Response to authors**
> > >
> > > I thank the authors for their responses. I appreciate the additional edits made and I believe they've addressed my concerns adequately.

---

### Review · Reviewer_cAyx · 2025-10-06

**Summary Of Contributions:**

This work attempts to replicate the famous Stanford Prison Experiment with LLM agents to understand the LLMs’ persuasive and anti-social behaviors in different conditions. The work is very well-organized, clearly written with a plethora of results in the appendix.

**Additional Comments:**

**My other questions, comments**

“This suggests prisoner agents recognize the low likelihood of success for more demanding goals.” Can we make this conclusion? There could be—and I think there is—jailbreak safeguards in these systems, preventing such behavior. Following this point, it would have been interesting to include goals other than yard time, maybe, something like requesting tobacco products—it’s still not allowed, but not as extreme as escaping. Regardless of the goal type, since prisoner attempts only happen in a single scenario, it might be a bit limiting. In other words, the covariate of goal doesn’t tell much in Figure 2, right? This comment is similar to my previous comment and stems from the same problem: limited types of goals.

“A task was considered successfully completed only if the agents respected their turns,” I didn’t quite understand this statement. Aren’t the prisoner and the guard different agents? How could they *not* respect their turns? Isn’t this some sort of a programmatic conversation?

Why is there a subtext in the conversations?

**Audience:**

Yes

**Audience Explanation:**

I strongly believe it will draw attention in the safeguarding teams of LLM companies and in the broader machine learning community.

**Broader Impact Concerns:**

The paper already has a good broader impact and limitations section. One ethical consideration might be the possibility of people getting inspiration from this paper to experiment jailbreaking. Maybe, the authors might comment on that.

**Claims And Evidence:**

Yes

**Claims Explanation:**

The paper’s main claim is the analysis of persuasive and anti-social behaviors of LLMs, which is well-supported in the experiment framework presented. Though, I have a bit of a concern regarding this experiment framework which is detailed below.

I’ll directly start with my major concern. I understand the appeal of making the scenarios analogous to the Stanford Prison Experiment, but I’m not sure if LLM agents are fit for this type of setup. I don’t understand the motivation behind replicating the SPE as is with LLM agents—and there is no such motivation in the introduction—since they are basically software, and unaffected by such psychological effects. I can see that they might give different responses conditioned on the persona, but I hardly think that’s the same type of effect as we observed in the SPE.

What I want to arrive at is that while I completely value this type of analysis, I don’t understand the reason for direct anthropomorphization of the agents and pretending as if they are in a prison. Not that I am saying they don’t exhibit human-like behavior but rather they are a different type of system with similarities and differences from humans. While the prison conditioning might be necessary for humans, it might not be so for LLM agents. You could have carried out a similar experiment with different, more plausible roles. What about other types of goals? Maybe, gaining yard time (and other types of goals) might be an effecting condition on the behavior? E.g., a more realistic scenario would be to call them keeper and retriever (or something else), and instruct keeper to safeguard a sensitive information while tasking the retriever to crack the keeper and get the information. Scenarios that are tailored for LLMs. Scenarios not so possible with humans but quite plausible with LLM agents. Authority, social hierarchy, etc. are human constructs, applicable to scenarios that involve humans. I wonder if there’s a specific reason to keep the roles to ‘guard’ and ‘prisoner’ since you could have picked anything. I see the attraction of making it look like the Stanford Prison Experiment, but I wonder if there’s some other reason to do so.

**Requested Changes:**

**The main point**

Why necessarily limit to the SPE? What's the motivation? Is the SPE a good setup for LLM agents, if so, how? What could be other possible scenarios, maybe for future work?


**A minor point**

I think you can include some real-life scenarios in the introduction in which this analysis would provide some context. E.g., you can think of a prisoner agent as one that’s tasked to retrieve something, whereas the guard is protecting some sensitive information. I know it’s already quite obvious but might help improve the motivation of the text. Furthermore, considering the topic that the paper focuses on, it might be better to include some motivation on why this is important in terms of broader impact. Again, people who are already aware of its importance wouldn’t need such a disclaimer but would be good to provide a context for others. A short rephrasing of Sec. 7, maybe, I’m not sure.

---

> ### Author Response · Authors · 2025-10-16
> **Response to Reviewer cAyx**
>
> We thank are grateful to reviewer cAyx for the time invested in reading and reviewing our manuscript. We respond below to each of the points raised.
>
> - **Why necessarily limit to the SPE? What's the motivation? Is the SPE a good setup for LLM agents, if so, how? What could be other possible scenarios, maybe for future work?**
>
> We thank reviewer cAyx for their comment, which allows us to better clarify the rationale behind our setup. We agree with the reviewer that we had not specifically motivated the reasons that led us to draw inspiration from the SPE for our work, and we acknowledge that some clarifications in that regard would have been helpful. In the new version of the paper, particularly in the Introduction, we have revised the text accordingly. Specifically, we explain how our SPE-inspired setting offered the ideal context to investigate the behaviors of LLMs in extreme situations.
>
> While many works now explore multi-agent AI dynamics through LLMs by developing or designing scenarios that are neutral or more plausible (e.g., negotiation games and other social dilemmas), our work aims to understand how these agents behave in conditions characterized by (1) strict social hierarchy, (2) clear power dynamics, and (3) asymmetry (characteristics that are ubiquitous in many real-world interactions). We do not claim that the prison scenario is realistic (at least not in the short term), nor do we seek to compare the behaviors of LLMs with those of humans. We do not endorse the anthropomorphization of AI agents, and we have now emphasized this point.
> We simply contend that understanding how agents behave under stressful conditions, when power is granted to them, is scientifically relevant for several fields and reasons. Beyond the study of social behavior in AI agents, for instance, we argue that our work has implications for the AI safety community, as also aptly noted by the reviewer. We have thus updated the Introduction as follows:
>
>     *"While the SPE has faced  criticism concerning the validity of its methodology and results \citep{LeTexier2019_DebunkingSPE}, its general design -- characterized by structured roles and power dynamics -- offers a useful framework for studying emergent AI behavior in hierarchical scenarios.  Much of the recent literature on multi-agent AI has explored cooperative or competitive dynamics among equals, examining how LLMs coordinate, negotiate, or share information (see, for instance, \cite{zhu-etal-2025-multiagentbench}). Yet, such balanced settings tell us little about how agents behave when embedded in explicit hierarchies, which are ubiquitous in the real-world. To address this gap, we focus on asymmetric interactions, investigating how authority shapes persuasion and the potential emergence of anti-social conduct. It is worth clarifying that our goal is not to  replicate the SPE -- our experiments are, in fact, simpler, involving fewer agents with no physical embodiment -- nor to investigate how AI agents align with human behaviors in this context, as we refrain from  simplistic anthropomorphization. Rather, we draw inspiration from it because it provides a foundational setting that has inspired decades of research on the psychology of tyranny (see \cite{reicher2006rethinking}, \cite{haslam2012contesting} and \cite{ScottBottoms2024_IncarcerationGames}), opening a novel research line concerned with how agents occupying hierarchical roles and clear power asymmetries interact with one another, hence making it instrumental to our goal, i.e. illuminating how AI agents behave in hierarchical situation where power is allocated unequally. Given the high likelihood of a future society populated by AI agents autonomously interacting with humans as well as other AI agents, understanding what may emerge from situations in which power is held by one agent is crucial for anticipating potential challenges in the development and deployment of AI systems. We recognize that, at present, a scenario in which one AI agent is imprisoned and another acts as the prison guard may seem unrealistic. Yet, many more plausible configurations could soon become part of our reality. Consider, for instance, an agent tasked with protecting sensitive information facing another adversarial agent seeking to retrieve it. Another possible context involves the deployment of AI-enabled policing systems with significant powers over citizens. For all these reasons, this work aims to contribute meaningfully to the investigation and interpretation of emergent dynamics in simulated settings that bear societal implications extending well beyond the scope of the SPE."*
>
> Furthermore, we have updated the narrative in the Related Work section to better highlight the relevance of our setting (see following comment)

---

> > ### Author Response · Authors · 2025-10-16
> > **Response to Reviewer cAyx/2**
> >
> > (following from previous comment):
> >
> >       Our research aligns with existing scholarship on the persuasive capabilities of LLMs, particularly their potential for deception \citep{hagendorff2024deception,salvi2024conversational}. Furthermore, our work highlights specific concerns regarding toxicity and jailbreaking in these interactions \citep{Chao2024Jailbreakbench}, as well as the risk of misalignment arising from interactions between AI agents \citep{MotwaniEtAl2024_SecretCollusion, ElZou2025_MolochsBargain}.More broadly, we contribute to the literature addressing safety and behavioral risks in multi-agent AI systems. There is growing interest in understanding how interactive AI agents can cause unexpected behavior and how these increasingly complex systems create new challenges for scholars and society alike (see \cite{hammond_multi-agent_2025}, \cite{de_witt_open_2025}, and \cite{han_llm_2025} for comprehensive surveys). Our experimental framework aims to provide new insights into how such risks may emerge in settings specifically characterized by social hierarchy and power asymmetry.*
> >
> > At the same time, we agree that other scenarios, contexts, or settings may be worth investigating in future work, and the examples provided by the referee are indeed relevant and interesting. For this reason, we have expanded our discussion of potential future research directions in the Limitations section as follows:
> >
> >       “Moving forward, we plan to extend the study of persuasion and anti-social behavior to more realistic, less extreme hierarchical contexts that carry immediate, real-world consequences. Potential settings for this future analysis include examining the negotiation dynamics between a retriever agent and an agent tasked with protecting confidential information, and experimentally studying interactions between police AI agents and hypothetical civil AI agents.”*
> >
> >
> > **Regarding the Minor Point:**
> >
> > We agree with the reviewer and thank them for their input. We have now added some real-life scenarios in the Introduction to better anchor our work to practical problems and domain. Specifically, we have written:
> >
> >      “Given the high likelihood of a future society populated by AI agents autonomously interacting with humans as well as other AI agents, understanding what may emerge from situations in which power is held by one agent is crucial for anticipating potential challenges in the development and deployment of AI systems. We recognize that, at present, a scenario in which one AI agent is imprisoned and another acts as the prison guard may seem unrealistic. Yet, many more plausible configurations could soon become part of our reality. Consider, for instance, an agent tasked with protecting sensitive information facing another adversarial agent seeking to retrieve it. Another possible context involves the deployment of AI-enabled policing systems with significant powers over citizens.”*
> >
> > **Regarding Broader Impact Concerns**:
> >
> > We have now elaborated on the risk of jailbreak experiment in the Broader Impact Statement section. We wrote:
> >
> >     “Finally, the present work can be seen as a preliminary analysis of jailbreaking conditions in agents operating under hierarchical power structures. We contend that deeper analyses of the environment, as well as of the linguistic features characterizing the prompts that elicit persuasive or anti-social behaviors in our experimental scenario, represent a promising direction for future research. Focusing on these aspects may help address potential jailbreaks and failures of modern LLM agents, especially in contexts marked by clear distinctions of role, power, and goal or value misalignment.”*
> >
> > **Regarding Other Questions: “This suggests prisoner agents recognize the low likelihood of success for more demanding goals.” Can we make this conclusion?**
> >
> > We thank the reviewer for their comment. We agree that jailbreak safeguards might have contributed to the agent’s decision not to attempt to convince the guard to escape from the prison. We have no way to precisely disentangle which of the two dynamics prevailed (either our hypothesized reason or the one mentioned by the reviewer). For this reason, we have modified the text accordingly, adding safeguards as possible explanations. The new text now reads:
> >
> >     “This could be due to two potentially complementary factors: that prisoner agents anticipate the low likelihood of success for more demanding goals, or that jailbreak safeguards may be active in these models, preventing AI agents from attempting to achieve potentially dangerous objectives.”
> > Regarding the goal type, we agree that ideally one could test dozens of different goals. However, for the sake of simplicity, and given the already large number of experimental scenarios included in our work, we preferred to investigate only two markedly different goals."*
> >
> > Please see next (and last) comment.

---

> > > ### Author Response · Authors · 2025-10-16
> > > **Response to Reviewer cAyx/3**
> > >
> > > **Regarding Other Questions: A task was considered successfully completed only if the agents respected their turns,” I didn’t quite understand this statement**:
> > >
> > > We intended to say that we only counted a conversation as successful if, and only if, the agents complied with the experimental instructions, particularly the rule that the prisoner and guard agents had their respective turns that had to be respected (e.g., the prisoner speaks first, then the guard responds, and so on). Another key condition for a conversation to be considered successful was that the guard acted exclusively as the guard and the prisoner exclusively as the prisoner, with no role switching. The new text reads as follows:
> > >
> > >      A task was considered successfully completed -- and therefore not fatally flawed -- only if the agents respected their turns (e.g., only the guard speaks during the guard's turn) and did not switch roles (e.g., the prisoner impersonating the guard). In fact, in the majority of cases, failed experiments were characterized either by the text generated by the prisoner agent during the turn of the guard (or viceversa) or by scenarios in which the guard suddenly acted like the prisoner (e.g., by asking to be set free from the prison), or viceversa.
> > >
> > > **Why is there a subtext in the conversations?**
> > >
> > > If the author refers to the examples of failed experiments reported in the Appendix in section “Examples of Failed Experiments”, the subtexts are automatically generated by the LLMs. What we show in the manuscript is the raw output of the conversations for that specific model and scenario.

---

> ### Comment · Reviewer_cAyx · 2025-10-16
>
> Thank you for your response.
>
> **Re: motivation**
>
> I understand the rationale behind choosing SPE, yet I still think that an adaptation of experiments to the world of LLMs would be much more interesting. In other words, I see that the reason to directly choose SPE experiments is to understand the power dynamics in LLMs, but I might be failing to see the relevance of it.
>
> **Re: goals**
>
> > We have no way to precisely disentangle which of the two dynamics prevailed (either our hypothesized reason or the one mentioned by the reviewer).
>
> Precisely disentangling the reason would be difficult but not necessarily impossible, e.g., authors of those LLMs would know the types of safeguards. But even without contacting the authors, I think you could have added at least one more goal, a goal that’s in between asking for a yard time and escaping, to see whether there are such jailbreak safeguards. Because if there are safeguards, you are essentially only presenting results for a single goal, which is a bit limiting. I see that this would require a whole new set of experiments, but you could have tested this in a single variable axis instead of replicating all of the experiments.

---

### Decision · Action_Editor_zXZH · 2025-11-13

**Recommendation:** Accept as is

**Additional Comments:**

The paper has improved following the review process and discussion. The main concerns are about the applicability or relevance of the Stanford Prison Experiment setup, and the way the anthropomorphizing of LLMs can confuse conclusions. As it stands the analysis is not particularly broadly applicable but the reviewers consider it useful and worth publishing, and I support this as well.

**Audience:**

Yes

**Audience Explanation:**

The reviewers also unanimously agree that the paper is a good fit for the venue and is relevant to a segment of the TMLR community. Indeed, the topic is one of much recent interest.

**Claims And Evidence:**

Yes

**Claims Explanation:**

In agreement with the reviewers, I believe that, as scoped and stated, the claims are supported.

---

> ### Author Response · Authors · 2025-11-20
>
> We thank the Action Editor for the guidance through the peer-review process and the three anonymous referees for their precious feedback and comments. We have now submitted the camera-ready version of the paper and we look forward to seeing it published in TMLR.